# Angiotensin II receptor blockers and oral squamous cell carcinoma survival: A propensity-score-matched cohort study

Ching-Nung Wu[1,2], Shao-Chun Wu[3], Wei-Chih Chen[1], Yao-Hsu Yang[4,5,6], Jo-Chi Chin[7], Chih-Yen Chien[1], Fu-Min Fang[8], Shau-Hsuan Li[9], Sheng-Dean Luo[1,10]☯*, Tai-Jan Chiu[9,10]☯*

**1** Department of Otolaryngology, Kaohsiung Chang Gung Memorial Hospital, and Chang Gung University College of Medicine, Kaohsiung, Taiwan, **2** Department of Public Health, College of Medicine, National Cheng Kung University, Tainan, Taiwan, **3** Department of Anesthesiology, Kaohsiung Chang Gung Memorial Hospital, and Chang Gung University College of Medicine, Kaohsiung, Taiwan, **4** Department of Traditional Chinese Medicine, Chang Gung Memorial Hospital, Chiayi, Taiwan, **5** Health Information and Epidemiology Laboratory of Chang Gung Memorial Hospital, Chiayi, Taiwan, **6** School of Traditional Chinese Medicine, College of Medicine, Chang Gung University, Taoyuan, Taiwan, **7** Department of Anesthesiology, Park One International Hospital, Kaohsiung, Taiwan, **8** Department of Radiation Oncology, Kaohsiung Chang Gung Memorial Hospital, and Chang Gung University College of Medicine, Kaohsiung, Taiwan, **9** Department of Hematology-Oncology, Kaohsiung Chang Gung Memorial Hospital, and Chang Gung University College of Medicine, Kaohsiung, Taiwan, **10** Graduate Institute of Clinical Medical Sciences, College of Medicine, Chang Gung University, Taoyuan, Taiwan

☯ These authors contributed equally to this work.
* kuerten@cgmh.org.tw (TJC); rsd0323@cgmh.org.tw (SDL)

**Data Availability Statement:** The data cannot be shared publicly because it is owned by Chang Gung Medical Branches and authors do not have permission to share the data. Data are available

## Abstract

### Objectives

Angiotensin II receptor blockers (ARBs) improve the survival rates of patients with various cancers. However, it remains unclear whether ARBs confer a survival benefit on patients with oral squamous cell carcinoma (OSCC). Here, we assessed the associations between ARB use and survival in patients with OSCC of different stages.

### Materials and methods

This was a 10-year retrospective cohort study of OSCC patients. We enrolled 7,558 patients diagnosed with oral cancer between January 2007 and December 2017 whose details had been entered into the Chang Gung Research Database. Seven hundred and fourteen patients were recruited from the Chang Gung Research Database after performing 1:1 propensity score-matching between ARB users and non-users. Cox's regression models with adjusted covariates were employed to detect factors influencing the survival rates of patients with OSCC.

### Results

Kaplan-Meier analysis revealed that the overall survival (OS) rate of 180-day ARB users increased ($p = 0.038$). Cox's regression models indicated that ARB use, younger patients, early-stage OSCC, and patients without diabetes mellitus were independently prognostic of

from the Department of Medical Research and Development for researchers who meet the criteria for access to confidential data. If researchers who are interested in the data, it could be requested and applied through the the following contact information. E-mail: taytay@cgmh.org.tw (Ms. Shu-Jyuan Chiou).

**Funding:** This research was funded by Kaohsiung Chang Gung Memorial Hospital, Taiwan, CFRPG8H0401 and CORPG8J0091 to CNW and SCW. The funders had no role in study design, data collection and analysis, decision to publish, or preparation of the manuscript.

**Competing interests:** The authors have declared that no competing interests exist.

improved OS. Increased OS was more prominent in 180-day ARB users in stage III, Iva, and IVb categories.

## Conclusions

ARB use for more than 180 days is associated with an increased survival rate and is a positive, independent prognostic factor in patients with OSCC. A further two-arm study should be conducted to confirm the clinical usefulness of ARBs in OSCC patients.

## Introduction

Oral cancer is one of the most frequently occurring cancers worldwide. Oral squamous cell carcinoma (OSCC) represents the most common type of oral cancer, constituting approximately 90% of all oral cancers [1]. In 2018, more than 355,000 individuals were diagnosed with oral cancer worldwide, and approximately 177,000 oral cancer-related deaths were reported [2]. Despite advances in surgical techniques and chemoradiotherapy, the prognosis of patients with OSCC remains unsatisfactory, especially for those diagnosed with advanced disease. Therefore, the identification of novel therapeutic targets in OSCC is of high clinical importance.

The renin-angiotensin system (RAS) is involved in the regulation of blood pressure. Therefore, angiotensin I-converting enzyme inhibitors (ACEIs) and angiotensin II type 1 receptor blockers (ARBs) are the most widely used anti-hypertensive drugs. A retrospective cohort study conducted by Lever and colleagues showed that the long-term ACEI use protected against cancer [3], suggesting that the local RAS played roles in tumor development and progression. Additionally, the RAS has been implicated in most human cancers; thus, the use of ACEIs/ARBs has been proposed as a promising anti-tumor strategy, which could potentially suppress tumor progression through inhibition of cancer cell proliferation and neovascularization [4]. Indeed, the combination of ACEIs/ARBs with conventional anti-cancer therapies has been shown to improve clinical outcomes of patients with various types of cancer, including breast, urothelial, and gastrointestinal tract cancers [5–8].

However, the clinical usefulness of RAS inhibitors in patients with OSCC remains unclear. Also, most previous studies did not separately evaluate the anti-neoplastic effects of ACEIs and ARBs; the drug classes were combined when exploring the clinical outcomes of cancer patients. The impacts of ARBs alone were inconsistent [9, 10], suggesting that the ACEIs exerted all of the observed anti-neoplastic effects. Thus, we investigated the efficacy of ARBs in patients with OSCC. We used the Chang Gung Memorial Hospital database to perform a 10-year, retrospective cohort study. Furthermore, we explored the effects of ARBs on patients with advanced-stage OSCC.

## Material and methods

### Study cohort

We enrolled 7558 patients diagnosed with oral cancer between January 2007 and December 2017 whose details had been entered into the Chang Gung Research Database [11–13]. Fig 1 is a flow chart of the cohort study design for statistical analysis. Patients with non-squamous cell carcinoma or unclear staging data were excluded. Patients who did not have surgery or distant metastasis were also excluded because survival in these two groups was markedly different

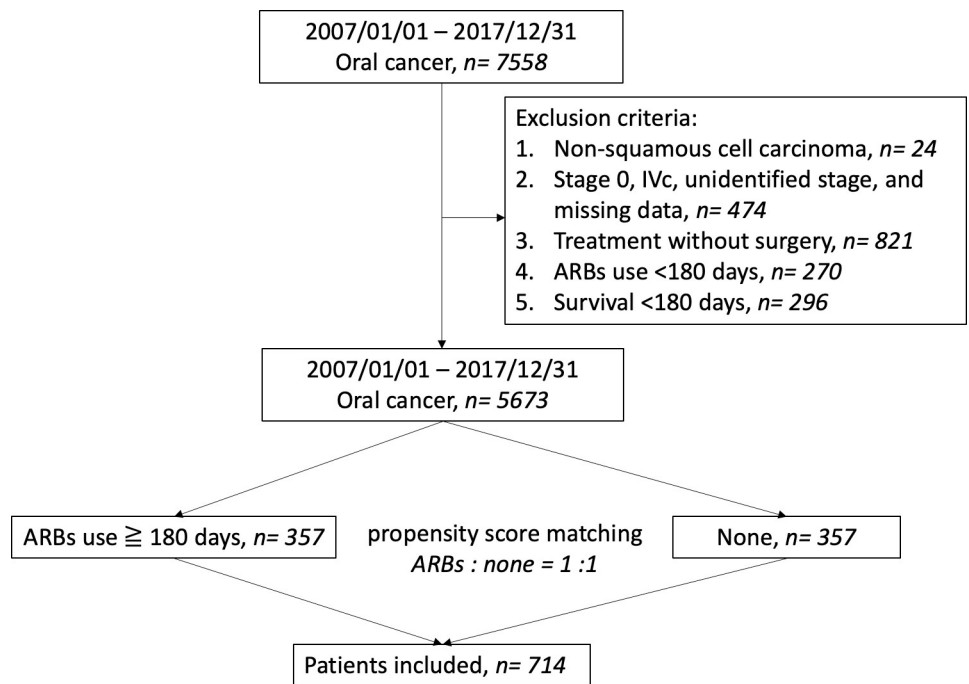

**Fig 1. Flow diagram illustrating propensity score matching in patients with oral cancer.** ARBs, angiotensin II receptor blockers.

from patients who received surgery [14]. Patients with ARB use < 180 days were excluded to prevent possible partial effects on survival. Non-ARB users with survival < 180 days were also excluded to make the two groups comparable. Hence, 5673 OSCC patients remained after applying the above exclusion criteria (Table 1). We performed propensity score-matching (PSM) to balance covariates between ARB users and non-users. Hence, data from 714 patients were analyzed in our study, including 357 patients treated with ARBs and 357 matched patients who did not receive ARBs (Table 2). This retrospective cohort study was approved by the Institutional Review Board (IRB) of Kaohsiung and Chiayi Chang Gung Memorial Hospital (Approval Nos. 202001463B0 and 201700253B0C602), and all experiments were performed in accordance with relevant guidelines and regulations. The Chang Gung Medical Foundation IRB (Approval No. 202001463B0) approved the waiver of participant consent.

## Statistical analyses

Categorical data (e.g., sex, comorbidities, lifestyle risk factors, cancer sites, and AJCC stage) were analyzed using a two-sided Fisher's exact test or a two-sided Pearson's chi-squared test. Parametric and non-parametric continuous data were analyzed using Student's *t*-test and the Mann–Whitney *U* test, respectively. To minimize the confounding effects due to non-randomized allocation, data were analyzed from a 1:1 propensity score-matched cohort (ARBs vs. nil), which had been identified by the Greedy method with a 0.25 caliper width using NCSS software, version 10 (NCSS Statistical Software, Kaysville, UT, USA). Propensity scores were calculated using a logistic regression model with sex, age, pathological AJCC stage, comorbidities, and the diagnostic year of OSCC as covariates (S1 Table). In ARB users, we calculated the survival time from the day of OSCC diagnosis if ABR was already used or from the day of starting ARB use if the patient had not used it after OSCC diagnosis. Because the diagnostic years were matched in both groups, the calculated survival time in non-users started from the same day as

**Table 1. Demographic and clinical characteristics of OSCC patients before matching.**

| Variables | OSCC patients n = 5673 | ARBs ≥ 180 days n = 362 | Non-Users n = 5311 | p value |
|---|---|---|---|---|
| **Median age at diagnosis**, years (IQR) | 52(45–60) | 58(51.7–66) | 52(45–59) | *<0.001 |
| **Gender** | | | | 0.501 |
| Female | 477(8.4%) | 27(7.5%) | 450(8.5%) | |
| Male | 5196(91.6%) | 335(92.5%) | 4861(91.5%) | |
| **Tumor sites** | | | | 0.352 |
| Lip | 292(5.1%) | 22(6.1%) | 270(5.1%) | |
| Oral tongue | 2105(37.1%) | 126(34.8%) | 1979(37.3%) | |
| Upper/lower Gum | 652(11.5%) | 51(14.1%) | 601(11.3%) | |
| Floor of mouth | 229(4.0%) | 12(3.3%) | 217(4.1%) | |
| Buccal mucosa | 1894(33.4%) | 125(34.5%) | 1769(33.3%) | |
| Hard palate | 86(1.5%) | 2(0.6%) | 84(1.6%) | |
| Retromolar trigone | 269(4.7%) | 18(5.0%) | 251(4.7%) | |
| Unidentified | 146(2.6%) | 6(1.7%) | 140(2.5%) | |
| **Lifestyle Risk Factors** | | | | |
| Smoking (n = 5647) | | | | *<0.001 |
| No | 1309(23.2%) | 112(30.9%) | 1197(22.6%) | |
| Yes | 4338(76.8%) | 250(69.1%) | 4088(77.4%) | |
| Betel nuts consumption (n = 5647) | | | | *0.001 |
| No | 2084(36.9%) | 162(44.8%) | 1922(36.4%) | |
| Yes | 3563(63.1%) | 200(55.2%) | 3363(63.6%) | |
| Alcoholic beverages (n = 5646) | | | | 0.194 |
| No | 2235(39.6%) | 155(42.8%) | 2080(39.4%) | |
| Yes | 3411(60.4%) | 207(57.2%) | 3204(60.6%) | |
| **Comorbidities** | | | | |
| Diabetes mellitus | | | | *<0.001 |
| No | 4816(84.9%) | 161(44.5%) | 4655(87.6%) | |
| Yes | 857(15.1%) | 201(55.5%) | 656(12.4%) | |
| Hypertension | | | | *<0.001 |
| No | 4614(81.3%) | 0(0.0%) | 4614(86.9%) | |
| Yes | 1059(18.7%) | 362(100.0%) | 697(13.1%) | |
| Hyperlipidemia | | | | *<0.001 |
| No | 5081(89.6%) | 174(48.1%) | 4907(92.4%) | |
| Yes | 592(10.4%) | 188(51.9%) | 404(07.6%) | |
| **Clinical AJCC 7[th] staging** | | | | *0.002 |
| I | 1324(23.3%) | 107(29.6%) | 1217(22.9%) | |
| II | 1436(25.3%) | 104(28.7%) | 1332(25.1%) | |
| III | 781(13.8%) | 42(11.6%) | 739(13.9%) | |
| IVa & IVb | 2132(37.6%) | 109(30.1%) | 2023(38.1%) | |
| **Pathological AJCC 7[th] staging** | | | | *<0.001 |
| I | 1555(27.4%) | 122(33.7%) | 1433(27.0%) | |
| II | 1318(23.2%) | 106(29.3%) | 1212(22.8%) | |
| III | 765(13.5%) | 45(12.4%) | 720(13.6%) | |
| IVa & IVb | 2035(35.9%) | 89(24.6%) | 1946(36.6%) | |
| **Treatment** | | | | *0.001 |
| Operation alone | 3108(54.8%) | 229(63.3%) | 2879(54.2%) | |
| Operation plus RT/CCRT | 2565(45.2%) | 133(36.7%) | 2432(45.8%) | |
| **Recurrence** | | | | *0.030 |

(*Continued*)

**Table 1.** (Continued)

| Variables | OSCC patients n = 5673 | ARBs ≥ 180 days n = 362 | Non-Users n = 5311 | p value |
|---|---|---|---|---|
| No | 4545(80.1%) | 306(84.5%) | 4239(79.8%) | |
| Yes | 1128(19.9%) | 56(15.5%) | 1072(20.2%) | |
| **Survival** | | | | *0.005 |
| Alive | 4135(72.9%) | 288(79.6%) | 3847(72.4%) | |
| Primary OSCC related death | 981(17.3%) | 41(11.3%) | 940(17.7%) | |
| Die of other reasons | 557(9.8%) | 33(9.1%) | 524(9.9%) | |

Abbreviations: AJCC, American Joint Committee on Cancer; ARBs, angiotensin II receptor blockers; CCRT, concurrent chemoradiotherapy; IQR, interquartile range; OSCC, oral squamous cell carcinoma; RT, radiotherapy.

its match to make the comparison between the two groups fair. The Kaplan–Meier method and log-rank test were used to evaluate the effects of ARBs on the primary outcome. The correlations between variables were evaluated using Pearson's correlation coefficient to prevent multicollinearity before building a regression model. Several models were built and tested as a sensitivity analysis, and the Cox proportional hazards model was used if the model met the criteria of the smallest Akaike information criterion (AIC). The Cox proportional hazards model tested the dependence of primary factors on other prognostic factors in multivariate survival modeling. Stratified analysis was performed and adjusted to analyze the efficacy of ARBs in patients at different pathological stages. All statistical analyses were performed using SAS software, version 9.4 of the SAS System for Windows (SAS Institute Inc., Cary, NC, USA). $P$-values $< 0.05$ were considered statistically significant.

## Results

Among the 7558 oral cancer patients, 5673 OSCC patients remained after applying the exclusion criteria (Table 1). In brief, age, lifestyle risk factors, and AJCC stages of cancer markedly differed between ARB users and non-users. In addition, there were significantly higher numbers of ARB users with comorbidities, such as hypertension, diabetes mellitus (DM), and hyperlipidemia, compared to non-users. Treatments were very different between the two groups as well.

A total of 714 patients were recruited for this cohort study after performing 1:1 PSM to balance covariates between the two groups, of which 357 were identified as ARB users and 357 as non-users after their diagnosis of cancer. Only ARB administration was significantly associated with hypertension. Otherwise, there were no statistically significant differences in clinical features between ARB-treated patients and those who did not receive ARB. Baseline clinico-pathological characteristics of the study cohort are summarized in Table 2. Of the 714 patients with OSCC, 92.3% (n = 659) were men and 7.7% (n = 55) were women. The median age at diagnosis was 58 years. Tongue (34.9%) and buccal mucosa (34.3%) were the most common tumor sites. In all, 443 patients (62.0%) had early-stage cancer (stage I or II), while the remaining 271 patients (38.0%) had advanced-stage tumors (stage III, IVA, or IVB). Also, 460 patients (64.4%) underwent surgery alone, and 254 (35.6%) underwent surgery plus adjuvant radiotherapy (RT) or concurrent chemoradiotherapy (CCRT). At the end of the study period, 165 (23.1%) patients had died; 94 of these (13.2%) had died of primary head and neck cancer.

Regarding the influence of prognostic factors on survival, univariate Cox regression analysis showed that various clinical variables, including age, pathological AJCC stages of cancer, treatments, DM, and ARB use, were significantly associated with overall survival (OS), while sex and hypertension were not significantly correlated with survival rate. On the other hand, only

**Table 2. Demographic and clinical characteristics of the study cohort.**

| Variables | Cohort n = 714 | ARBs ≥ 180 days n = 357 | Non-Users n = 357 | p value |
|---|---|---|---|---|
| **Median age at diagnosis**, years (IQR) | 58(52–66) | 58(51–66) | 59(52–66) | 0.426 |
| **Gender** | | | | 0.888 |
| Female | 55(7.7%) | 27(7.6%) | 28(7.8%) | |
| Male | 659(92.3%) | 330(92.4%) | 329(92.2%) | |
| **Tumor sites** | | | | 0.932 |
| Lip | 43(6.0%) | 21(5.9%) | 22(6.2%) | |
| Oral tongue | 249(34.9%) | 125(35.0%) | 124(34.7%) | |
| Upper/lower Gum | 99(13.9%) | 50(14.0%) | 49(13.7%) | |
| Floor of mouth | 21(2.9%) | 12(3.4%) | 9(2.5%) | |
| Buccal mucosa | 245(34.3%) | 124(34.7%) | 121(33.9%) | |
| Hard palate | 5(0.7%) | 2(0.6%) | 3(0.8%) | |
| Retromolar trigone | 37(5.2%) | 18(5.0%) | 19(5.3%) | |
| Unidentified | 15(2.1%) | 5(1.4%) | 10(2.8%) | |
| **Lifestyle Risk Factors** | | | | |
| Smoking | | | | 0.324 |
| No | 210(29.4%) | 111(31.1%) | 99(27.7%) | |
| Yes | 504(70.6%) | 246(68.9%) | 258(72.3%) | |
| Betel nuts consumption | | | | 0.598 |
| No | 315(44.1%) | 161(45.1%) | 154(43.1%) | |
| Yes | 399(55.9%) | 196(54.9%) | 203(56.9%) | |
| Alcoholic beverages | | | | 0.084 |
| No | 329(46.1%) | 153(42.9%) | 176(49.3%) | |
| Yes | 385(53.9%) | 204(57.1%) | 181(50.7%) | |
| **Comorbidities** | | | | |
| Diabetes mellitus | | | | 0.598 |
| No | 315(44.1%) | 161(45.1%) | 154(43.1%) | |
| Yes | 399(55.9%) | 196(54.9%) | 203(56.9%) | |
| Hypertension | | | | *<0.001 |
| No | 223(31.2%) | 0(0.0%) | 223(62.5%) | |
| Yes | 491(68.8%) | 357(100.0%) | 134(37.5%) | |
| Hyperlipidemia | | | | 0.369 |
| No | 360(50.4%) | 174(48.7%) | 186(52.1%) | |
| Yes | 354(49.6%) | 183(51.3%) | 171(47.9%) | |
| **Clinical AJCC 7th staging** | | | | 0.110 |
| I | 192(26.9%) | 105(29.4%) | 87(24.4%) | |
| II | 215(30.1%) | 101(28.3%) | 114(31.9%) | |
| III | 101(14.1%) | 42(11.8%) | 59(16.5%) | |
| IVa & IVb | 206(28.9%) | 109(30.5%) | 97(27.2%) | |
| **Pathological AJCC 7th staging** | | | | 0.927 |
| I | 231(32.4%) | 119(33.3%) | 112(31.4%) | |
| II | 212(29.7%) | 104(29.1%) | 108(30.3%) | |
| III | 94(13.2%) | 45(12.6%) | 49(13.7%) | |
| IVa & IVb | 177(24.8%) | 89(24.9%) | 88(24.6%) | |
| **Treatment** | | | | 0.348 |
| Operation alone | 460(64.4%) | 224(62.7%) | 236(66.1%) | |
| Operation plus RT/CCRT | 254(35.6%) | 133(37.3%) | 121(33.9%) | |
| **Recurrence** | | | | 0.758 |

(*Continued*)

**Table 2.** (Continued)

| Variables | Cohort n = 714 | ARBs ≥ 180 days n = 357 | Non-Users n = 357 | p value |
|---|---|---|---|---|
| No | 601(84.2%) | 302(84.6%) | 299(83.8%) | |
| Yes | 113(15.8%) | 55(15.4%) | 58(16.2%) | |
| **Survival** | | | | 0.176 |
| Alive | 549(76.9%) | 285(79.8%) | 264(73.9%) | |
| Primary OSCC related death | 94(13.2%) | 41(11.5%) | 53(14.8%) | |
| Die of other reasons | 71(09.9%) | 31(08.7%) | 40(11.2%) | |

Abbreviations: AJCC, American Joint Committee on Cancer; ARBs, angiotensin II receptor blockers; CCRT, concurrent chemoradiotherapy; IQR, interquartile range; OSCC, oral squamous cell carcinoma; RT, radiotherapy.

pathological AJCC stages of cancer and treatments were statistically significantly associated with disease-specific survival (DSS) (Table 3). In this 10-year cohort study, patients receiving ARBs for more than 180 days exhibited a significantly higher OS rate than those who did not receive ARBs (Fig 2). However, the DSS rate was not statistically significantly different between patients receiving ARBs and those not receiving ARBs (Fig 3).

Several regression models were built before a final Cox regression model could be determined (Table 4). Model 1 was adjusted for all potential confounders, including age, sex, pathological AJCC 7th staging, treatment, and diabetes mellitus. Model 2 was adjusted for suspected confounders according to the crude associations in univariate analyses. Model 3 was built according to the stepwise solution in statistical software. The stable effect size was found across different models in either OS or DSS. The final Cox proportional hazards models were chosen for OS and DSS if the models met the criteria of the smallest AIC. Therefore, models 2 and 3 were selected for OS and DSS, respectively.

According to the chosen models, multivariate analyses revealed that only advanced disease was associated with reduced OS and DSS. In addition, aging and diabetes mellitus were related to poor OS. Notably, 180-day ARB use was associated with improved OS (HR$_{\text{ARB users vs. non-users}}$ = 0.73, 95% CI = 0.53–0.99) but was not statistically significantly associated with improved DSS (HR$_{\text{ARB users vs. non-users}}$ = 0.73, 95% CI = 0.48–1.10) in patients with resectable oral cancer (Table 5). Furthermore, a survival benefit with at least 180 days of ARB use was observed in patients with stages III and IV OSCC (HR$_{\text{ARB users vs. non-users}}$ = 0.61, 95% CI = 0.39–0.94), but no statistical significance was observed in patients with stages I and II OSCC in advanced analysis (Table 6). Overall, these analyses suggest that patients with late-stage OSCC are the most likely to benefit from ARB use for more than 180 days after OSCC diagnosis.

## Discussion

To the best of our knowledge, this was the first study to investigate the potential clinical benefit of ARBs in patients with OSCC receiving surgery. In this 10-year retrospective cohort study, patients who received ARBs for at least 180 days had improved OS compared to patients who did not receive ARBs. In addition, patients with locally advanced OSCC experienced the most significant benefit from ARBs.

The RAS consists of several enzymatic and non-enzymatic protein components and is essential for the maintenance of vascular homeostasis. Angiotensinogen is produced in the liver and cleaved by the aspartyl protease renin to angiotensin I. Angiotensin I is subsequently cleaved by the angiotensin I-converting enzyme to produce angiotensin II (Ang II). Ang II is a key component of the RAS, which exerts its actions by binding to two G protein-coupled receptors: angiotensin receptor 1 (AT1R) and the lesser known angiotensin receptor 2 [15]. It

**Table 3. Univariate analyses of prognostic factors for OS and DSS in patients with oral cancer.**

| Factor | Cohort | OS | | DSS | |
|---|---|---|---|---|---|
| | | Hazard ratio (95% CI) | p value | Hazard ratio (95% CI) | p value |
| Age (year (IQR)) | 58(52–66) | 1.04(1.02–1.05) | *<0.001 | 1.01(0.99–1.03) | 0.403 |
| Gender | | | 0.707 | | 0.822 |
| Female | 55(07.7%) | 1 | | 1 | |
| Male | 659(92.3%) | 0.90(0.53–1.54) | | 0.92(0.45–1.90) | |
| Pathological AJCC 7th staging | | | *<0.001 | | *<0.001 |
| I | 231(32.4%) | 1 | | 1 | |
| II | 212(29.7%) | 1.06(0.69–1.65) | | 1.37(0.72–2.60) | |
| III | 94(13.2%) | 1.22(0.71–2.10) | | 1.49(0.68–3.26) | |
| IVa & IVb | 177(24.8%) | 2.83(1.91–4.18) | | 4.57(2.62–7.98) | |
| Treatment | | | *<0.001 | | *<0.001 |
| Operation alone | 460(64.4%) | 1 | | 1 | |
| Operation plus RT/CRT | 254(35.6%) | 1.85(1.36–2.51) | | 2.58(1.72–3.87) | |
| Diabetes mellitus | | | *0.011 | | 0.175 |
| No | 315(44.1%) | 1 | | 1 | |
| Yes | 399(55.9%) | 1.52(1.10–2.10) | | 1.34(0.88–2.03) | |
| Hypertension | | | 0.097 | | 0.207 |
| No | 223(31.2%) | 1 | | 1 | |
| Yes | 491(68.8%) | 0.76(0.56–1.05) | | 0.76(0.50–1.16) | |
| ARBs use | | | *0.038 | | 0.121 |
| No | 357(50.0%) | 1 | | 1 | |
| ≥ 180 days | 357(50.0%) | 0.72(0.53–0.98) | | 0.72(0.48–1.09) | |

Abbreviations: AJCC, American Joint Committee on Cancer; ARBs, angiotensin II receptor blockers; CRT, chemoradiotherapy; DSS, disease specific survival; OS, overall survival; RT, radiotherapy.

* $p \leq 0.05$.

is increasingly evident that, in addition to systemic effects on blood pressure and fluid homeostasis, AT1R and Ang II have important roles at the local tissue level. AT1R overexpression has been reported in numerous cancers, including ovarian, breast, and bladder cancer [16, 17]. Consistent with these findings, ARBs and ACEIs have been reported to reduce tumor growth and vascularization in a wide range of cancers, suggesting a role for Ang II in cancer development and progression [4]. We summarized the studies containing ARBs in the last ten years and revealed the effects on OS and DSS across different malignancies (Table 7) [5–8, 18–23]. However, there was no consistent conclusion on whether ARBs have a survival benefit in patients coexisting with malignancies. One of the main reasons was the discrepancy in confounding control across these studies. Unknown confounding and selection bias might exist in these retrospective studies with different study design, indicating the findings are only the association between ARBs use and survival. Future two-arm controlled study should be conducted for strengthening the causal relationship. In addition, the varying extent of concurrent ACEI and ARB use among previous studies may confound their effects. Therefore, we evaluated patients who had received ARBs only in our study (S2 Table).

Head and neck cancer is the sixth most common cancer worldwide [24]. The critical role of RAS in head and neck cancer has been shown in various tissues, including the oral mucosa [25]. Additionally, Ang II also has been found to promote HNSCC cell migration and invasion [26]. The effects of Ang II on autocrine and paracrine signaling pathways are mediated by AT1R, suggesting that ARBs might provide a clinical benefit in patients with HNSCC. This was approved

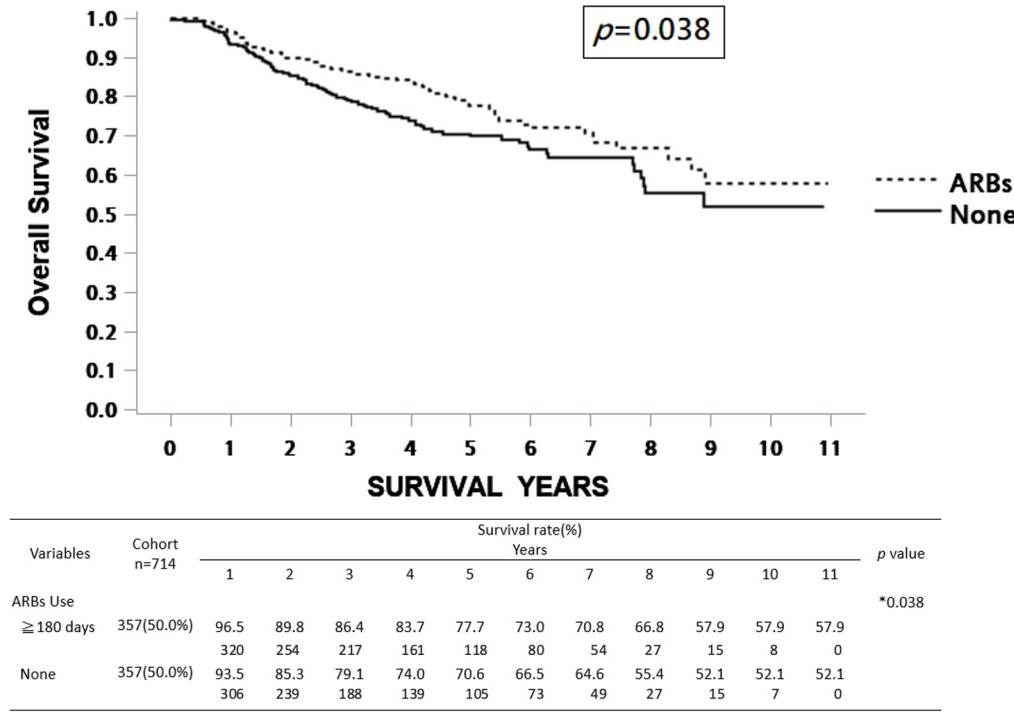

| Variables | Cohort n=714 | Survival rate(%) Years | | | | | | | | | | | p value |
|---|---|---|---|---|---|---|---|---|---|---|---|---|---|
| | | 1 | 2 | 3 | 4 | 5 | 6 | 7 | 8 | 9 | 10 | 11 | |
| ARBs Use | | | | | | | | | | | | | *0.038 |
| ≧180 days | 357(50.0%) | 96.5 | 89.8 | 86.4 | 83.7 | 77.7 | 73.0 | 70.8 | 66.8 | 57.9 | 57.9 | 57.9 | |
| | | 320 | 254 | 217 | 161 | 118 | 80 | 54 | 27 | 15 | 8 | 0 | |
| None | 357(50.0%) | 93.5 | 85.3 | 79.1 | 74.0 | 70.6 | 66.5 | 64.6 | 55.4 | 52.1 | 52.1 | 52.1 | |
| | | 306 | 239 | 188 | 139 | 105 | 73 | 49 | 27 | 15 | 7 | 0 | |

**Fig 2. Kaplan–Meier survival curve of OS rates between ARBs users (≥180 days) and non-users.** The estimated 5- and 10-year OS rates of ARB non-users (None) were 70.6% and 52.1%, respectively. The estimated 5- and 10-year OS rates of ARB users (≥180 days) were 77.7% and 57.9%, respectively. ARBs, angiotensin II receptor blocker; OS, overall survival.

by Lin et al., which ARBs were found to exert antiproliferative and antiangiogenesis effects by inducing apoptosis in nasopharyngeal carcinoma (NPC). In addition, improved 5-year OS and DSS were found among patients with NPC using ARBs [27]. Interestingly, we found that ARB use for at least 180 days improved the OS rate of patients with OSCC statistically, yet not DSS rate (Table 5). Accordingly, RAS is a major regulator of blood pressure (BP) and vascular response to injury. There is large evidence that RAS inhibition provides end-organ protection independent of BP lowering [28]. That probably explained the survival benefit of using ARBs is mainly through as end-organ protective effect, which further reduced overall mortality [29]. As for the anti-cancer effect of ARBs in oral cancer patients, it still remained controversial in our current study. Notably, the survival effects were most pronounced for patients with late-stage resectable OSCC (Table 6), suggesting that pathological staging served as an effect modifier.

We included ARB users for at least 180 days, as these medications are unlikely to have immediate effects on cancer progression [20]. In addition, the best lag-time to be applied in studies accordingly was around 6 months, which was the most appropriate period for the assessment of drug exposure [30]. On the other hand, we excluded patients without ARB use surviving < 180 days after the index date to make the comparison between the two groups fair. Although ARB use was not associated with a statistically improved DSS rate in our study, the hazard rate was similar to OS after adjusting for potential confounding factors (Table 4). The possible reason for the lack of statistical power was that our sample size was not large enough. The use of PSM to reduce the bias due to confounding variables was the highlight of our study, although the matched sample size was limited. When we used a 1:2 or larger ratio of matching, unmatched cases dramatically increased. Therefore, we created 1:1 PSM in our study, which only dropped five patients in the group of ARB users (S1 Table). As major comorbidity, hypertension was not

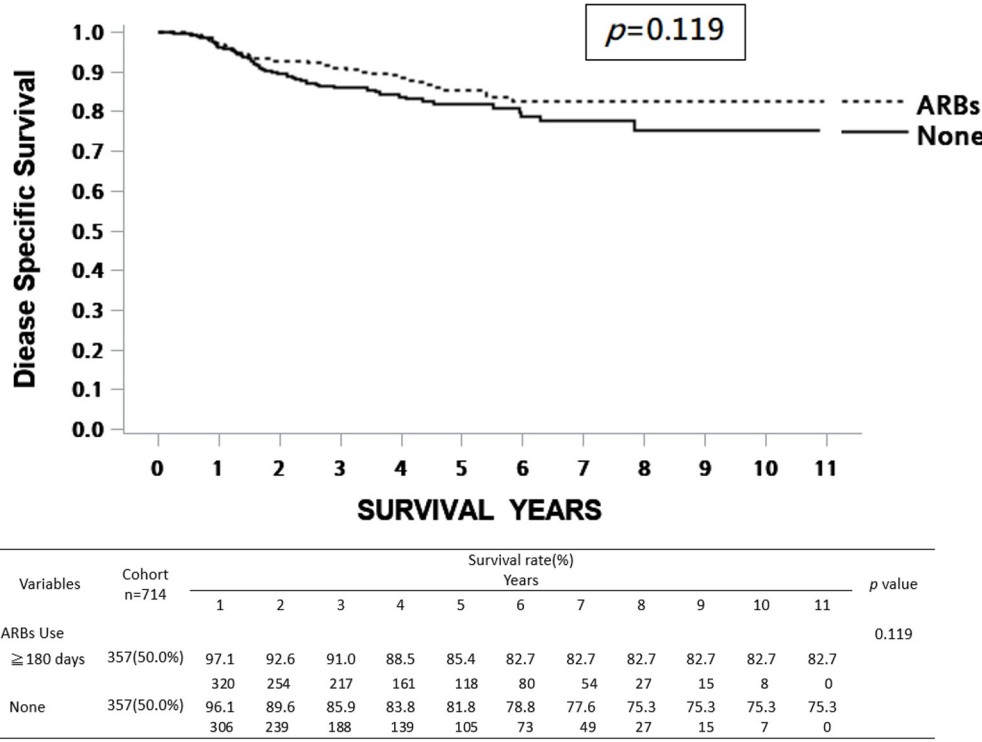

| Variables | Cohort n=714 | Survival rate(%) Years | | | | | | | | | | | p value |
|---|---|---|---|---|---|---|---|---|---|---|---|---|---|
| | | 1 | 2 | 3 | 4 | 5 | 6 | 7 | 8 | 9 | 10 | 11 | |
| ARBs Use | | | | | | | | | | | | | 0.119 |
| ≧180 days | 357(50.0%) | 97.1 | 92.6 | 91.0 | 88.5 | 85.4 | 82.7 | 82.7 | 82.7 | 82.7 | 82.7 | 82.7 | |
| | | 320 | 254 | 217 | 161 | 118 | 80 | 54 | 27 | 15 | 8 | 0 | |
| None | 357(50.0%) | 96.1 | 89.6 | 85.9 | 83.8 | 81.8 | 78.8 | 77.6 | 75.3 | 75.3 | 75.3 | 75.3 | |
| | | 306 | 239 | 188 | 139 | 105 | 73 | 49 | 27 | 15 | 7 | 0 | |

**Fig 3. Kaplan–Meier survival curve of DSS rates between ARB users (≥180 days) and non-users.** The estimated 5- and 10-year DSS rates of ARB non-users (None) were 81.8% and 75.3%, respectively. The estimated 5- and 10-year DSS rates of ARB users (≥180 days) were 85.4% and 82.7%, respectively. ARBs, angiotensin II receptor blockers; DSS, disease-specific survival.

selected to be a matched variable in our study because of its high correlation with ARB users (Pearson's correlation coefficient = 0.673, $p<0.001$). If we input hypertension into the PSM model, the matched non-ARB users would be highly associated with hypertension. The matched candidates would be scarce, and the generalizability would be limited.

Our study had a few limitations. First, medical records were incomplete for some patients. Therefore, some critical clinicopathological characteristics (e.g., surgical margin, extranodal extension, and depth of tumor invasion) could not be analyzed in our study. Second, although

**Table 4. Modeling for the effects of ARBs on OS and DSS in patients with resectable OSCC.**

| Outcomes | ARBs ≥ 180 days (n = 357) | Non-Users (n = 357) | Adjusted Hazard Ratio (95% CI) | | |
|---|---|---|---|---|---|
| | | | *Model 1 | **Model 2 | ***Model 3 |
| OS | 79.8% | 73.9% | 0.73(0.53–0.99) | 0.73(0.53–0.99) | 0.74(0.54–1.01) |
| | | | AIC = 1860.9 | AIC = 1858.9 | AIC = 1864.7 |
| DSS | 88.5% | 85.2% | 0.72(0.48–1.09) | 0.71(0.47–1.07) | 0.73(0.48–1.10) |
| | | | AIC = 1106.0 | AIC = 1107.2 | AIC = 1103.5 |

Abbreviations: AIC, Akaike information criterion; ARBs, angiotensin II receptor blockers; DSS, disease-specific survival; OS, overall survival; OSCC, oral squamous cell carcinoma.

*Model 1 was adjusted for all potential confounders, including age, sex, pathological AJCC 7th staging, treatment, and diabetes mellitus.

**Model 2 was adjusted for suspected confounders according to the crude associations in Table 3. For OS, age, pathological AJCC 7th staging, treatment, and diabetes mellitus were adjusted; for DSS, pathological AJCC 7th staging and treatment were adjusted.

***Model 3 was built with age, pathological AJCC 7th staging, and diabetes mellitus adjustment according to the statistical software (stepwise solution).

**Table 5. Multivariate analyses of prognostic factors for OS and DSS in patients with oral cancer.**

| Factor | Cohort | **OS Hazard ratio (95% CI) | p value | ***DSS Hazard ratio (95% CI) | p value |
|---|---|---|---|---|---|
| Age (year (IQR)) | 58(52–66) | 1.05(1.03–1.06) | *<0.001 | 1.02(0.99–1.04) | 0.058 |
| Pathological AJCC 7th staging | | | *<0.001 | | *<0.001 |
| I | 231(32.4%) | 1 | | 1 | |
| II | 212(29.7%) | 1.09(0.70–1.71) | | 1.40(0.73–2.65) | |
| III | 94(13.2%) | 1.19(0.66–2.13) | | 1.50(0.68–3.30) | |
| IVa & IVb | 177(24.8%) | 3.21(1.93–5.34) | | 5.08(2.88–8.95) | |
| Treatment | | | 0.355 | --- | --- |
| Operation alone | 460(64.4%) | 1 | | | |
| Operation plus RT/CRT | 254(35.6%) | 1.21(0.80–1.84) | | | |
| Diabetes mellitus | | | *0.003 | | 0.113 |
| No | 315(44.1%) | 1 | | 1 | |
| Yes | 399(55.9%) | 1.63(1.17–2.25) | | 1.40(0.92–2.14) | |
| ARBs use | | | *0.049 | | 0.142 |
| No | 357(50.0%) | 1 | | 1 | |
| ≥ 180 days | 357(50.0%) | 0.73(0.53–0.99) | | 0.73(0.48–1.10) | |

Abbreviations: AJCC, American Joint Committee on Cancer; ARBs, angiotensin II receptor blockers; CRT, chemoradiotherapy; DSS, disease specific survival; OS, overall survival; RT, radiotherapy.

* $p \le 0.05$.

** Model for OS was adjusted for age, pathological AJCC 7th staging, treatment and diabetes mellitus according to the smallest AIC in Table 4.

** Model for DSS was adjusted for age, pathological AJCC 7th staging and diabetes mellitus according to the smallest AIC in Table 4.

our study included only data from patients treated with ARBs, a considerable proportion of the patients were receiving concurrent treatment with other agents to control hypertension, which might have influenced our findings. Several patients received amlodipine and hydrochlorothiazide (S2 Table); however, there is no evidence that these agents suppress cancer development or progression [31, 32]. Therefore, we presume that treatment with these agents had a minimal influence on our findings. Third, various ARBs were included; therefore, the standardized effective dosage was difficult to calculate, and the dose-response relationship could not be measured in our study. Lastly, blood pressure, a crucial variate in an anti-hypertensive medication study, was not taken into account in our research. The main reasons were that complete data of blood pressure was not available in our database, and many confounding factors existed in the residual database as well. However, taking representative blood pressure data into analysis should be considered in future studies. In conclusion, a future prospective controlled study should be conducted to overcome these limitations.

**Table 6. Effects of ARBs on OS in patients with early and advanced OSCC.**

| Pathological AJCC staging | Variables | Death | Alive | Crude HR (95% CI) | p-value | **Adjusted HR (95% CI) | p-value |
|---|---|---|---|---|---|---|---|
| Stage I & II | **None** | 41(18.6%) | 179(81.4%) | 1 | 0.619 | 1 | 0.684 |
| | **ARBs** | 39(17.5%) | 184(82.5%) | 0.90(0.58–1.39) | | 0.92(0.59–1.42) | |
| Stage III & IV | **None** | 52(38.0%) | 85(62.0%) | 1 | *0.018 | 1 | *0.026 |
| | **ARBs** | 33(24.6%) | 101(75.4%) | 0.59(0.38–0.91) | | 0.61(0.39–0.94) | |

Abbreviations: AJCC, American Joint Committee on Cancer; ARBs, angiotensin II receptor blockers; HR, Hazard Ratio; OSCC, oral squamous cell carcinoma.

* $p \le 0.05$.

** Model was adjusted for age, treatment and diabetes mellitus.

**Table 7. Summary of previous research on the effects of ARBs on OS and DSS across different malignancies.**

| Cancer | Studies | Medication | Sample size | Outcomes | Study design | Notes |
|---|---|---|---|---|---|---|
| Breast | Holmes et al. (2013) | ACEi/ARBs | Exp: 880 Non-E: 2310 | OS aHR (95% CI): **1.22 (1.04, 1.44)** | Retrospective cohort | Not control comorbidities |
| | Cardwell et al. (2014) | ARBs | Cases: 648 Controls: 3193 | OS aOR (95% CI): **0.79 (0.60, 1.03)** DSS aOR (95% CI): **0.94 (0.65, 1.37)** | Nested case–control | --- |
| Esophageal | Busby et al. (2018) | ARBs | Exp: 168 Non-E: 2565 | DSS aHR (95% CI): **0.89 (0.71, 1.10)** | Retrospective cohort | Not control stages |
| Gastric | Kim et al. (2012) | ACEi/ARBs | Exp: 30 Non-E: 33 | OS aHR (95% CI): **0.54 (0.30, 0.97)** | Retrospective cohort | Advanced stages |
| | Busby et al. (2018) | ARBs | Exp: 168 Non-E: 2565 | DSS aHR (95% CI): **0.79 (0.62, 1.00)** | Retrospective cohort | Not control stages |
| Liver | Facciorusso et al. (2015) | ARBs | Exp: 43 Non-E: 113 | OS HR (95% CI): **0.71 (0.46–1.10)** | Retrospective cohort | Not adjust for HR |
| Pancreas | Nakai et al. (2010) | ACEi/ARBs | Exp: 27 Non-E: 103 | OS aHR (95% CI): **0.52 (0.29, 0.88)** | Retrospective cohort | --- |
| | Cerullo et al. (2017) | ARBs | Exp: 479 Non-E: 3820 | OS aHR (95% CI): **0.76 (0.67, 0.87)** | Retrospective cohort | --- |
| Colorectal | Holmes et al. (2013) | ACEi/ARBs | Exp: 1187 Non-E: 1864 | OS aHR (95% CI): **1.03 (0.93, 1.15)** | Retrospective cohort | Not control comorbidities |
| | Cardwell et al. (2014) | ARBs | Cases: 1093 Controls: 5231 | OS aOR (95% CI): **1.02 (0.82, 1.26)** DSS aOR (95% CI): **0.80 (0.59, 1.09)** | Nested case–control | --- |
| | Morris et al. (2016) | ACEi/ARBs | Exp: 25 Non-E: 90 | OS aOR (95% CI): **0.73 (0.45–1.20)** | Retrospective cohort | --- |
| Renal | Asgharzadeh et al. (2020) | ARBs | --- | OS aHR (95% CI): **0.81 (0.69, 0.96)** | Meta-analysis | --- |
| Prostate | Cardwell et al. (2014) | ARBs | Cases: 766 Controls: 3777 | OS aOR (95% CI): **0.92 (0.75, 1.12)** DSS aOR (95% CI): **0.82 (0.61, 1.11)** | Nested case–control | --- |
| | Mao et al. (2016) | ACEi/ARBs | --- | OS aRR (95% CI): **0.92 (0.87, 0.98)** | Meta-analysis | --- |

Abbreviations: ACEi, Angiotensin converting enzyme inhibitors; aHR, adjusted Hazard ratio; aOR, adjusted Odds ratio; aRR, adjusted risk ratio; ARBs, angiotensin II receptor blockers; DSS, disease specific survival; Exp, exposure; HR, Hazard ratio; Non-E, non-exposure; OS, overall survival; OSCC, oral squamous cell carcinoma.

## Conclusion

This was the first study to investigate the clinical usefulness of ARBs in patients with OSCC receiving surgery. Patients who received ARBs for at least 180 days exhibited improved OS. Additionally, ARBs use was associated with a more significant survival benefit in patients with operable stage III, IVa, and IVb OSCC. A further two-arm study should be conducted to confirm the clinical usefulness of ARBs in OSCC patients.

## Supporting information

**S1 Table. Baseline characteristics of OSCC patients before and after propensity-score matching.**
(DOCX)

**S2 Table. The included angiotensin II receptor blockers and numbers in our study.**
(DOCX)

## Acknowledgments

We thank the Health Information and Epidemiology Laboratory at the Chiayi Chang Gung Memorial Hospital and the Biostatistics Center at Kaohsiung Chang Gung Memorial Hospital for statistics work.

## Author Contributions

**Conceptualization:** Ching-Nung Wu, Tai-Jan Chiu.

**Data curation:** Yao-Hsu Yang, Fu-Min Fang.

**Formal analysis:** Shao-Chun Wu, Yao-Hsu Yang, Jo-Chi Chin.

**Funding acquisition:** Tai-Jan Chiu.

**Investigation:** Wei-Chih Chen.

**Methodology:** Ching-Nung Wu.

**Project administration:** Tai-Jan Chiu.

**Resources:** Wei-Chih Chen, Shau-Hsuan Li.

**Software:** Shao-Chun Wu, Jo-Chi Chin.

**Supervision:** Sheng-Dean Luo.

**Validation:** Shao-Chun Wu.

**Visualization:** Chih-Yen Chien, Sheng-Dean Luo.

**Writing – original draft:** Ching-Nung Wu.

**Writing – review & editing:** Sheng-Dean Luo.

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
