## [Decision Letter · Decision Letter 0]

14 Sep 2021

PONE-D-21-16892Angiotensin II receptor blockers and oral squamous cell carcinoma survival: A propensity-score-matched cohort studyPLOS ONE

Dear Dr. Chiu,

Thank you for submitting your manuscript to PLOS ONE. After careful consideration, we feel that it has merit but does not fully meet PLOS ONE’s publication criteria as it currently stands. Therefore, we invite you to submit a revised version of the manuscript that addresses the points raised during the review process.

Two Reviewers well assessed this manuscript.  However, several major revisions are needed in the present form.  See the Reviewers’ comments and respond them appropriately.

We look forward to receiving your revised manuscript.

Kind regards,

Masaki Mogi

Academic Editor

PLOS ONE

2. Thank you for stating the following in the Acknowledgments/Funding Section of your manuscript:

“This research was funded by Kaohsiung Chang Gung Memorial Hospital, Taiwan, CFRPG8H0401 and CORPG8J0091”

“This research was funded by Kaohsiung Chang Gung Memorial Hospital, Taiwan, CFRPG8H0401 and CORPG8J0091 to CNW and SCW. The funders had no role in study design, data collection and analysis, decision to publish, or preparation of the manuscript.”

4. PLOS requires an ORCID iD for the corresponding author in Editorial Manager on papers submitted after December 6th, 2016. Please ensure that you have an ORCID iD and that it is validated in Editorial Manager. To do this, go to ‘Update my Information’ (in the upper left-hand corner of the main menu), and click on the Fetch/Validate link next to the ORCID field. This will take you to the ORCID site and allow you to create a new iD or authenticate a pre-existing iD in Editorial Manager. Please see the following video for instructions on linking an ORCID iD to your Editorial Manager account: https://www.youtube.com/watch?v=_xcclfuvtxQ.

Additional Editor Comments (if provided):

Reviewers' comments:

Reviewer's Responses to Questions

**Comments to the Author**

1. Is the manuscript technically sound, and do the data support the conclusions?

Reviewer #1: Partly

Reviewer #2: Yes

2. Has the statistical analysis been performed appropriately and rigorously? 

Reviewer #1: Yes

Reviewer #2: Yes

3. Have the authors made all data underlying the findings in their manuscript fully available?

Reviewer #1: Yes

Reviewer #2: Yes

4. Is the manuscript presented in an intelligible fashion and written in standard English?

Reviewer #1: Yes

Reviewer #2: Yes

5. Review Comments to the Author

Reviewer #1: The authors reported that ARB use for more than 180 days is associated with an increased survival rate and is a positive, independent prognostic factor in patients with oral squamous cell carcinoma(OSCC) in their propensity-score-matched cohort from registry database.

It is interesting, but also arguable results. As the authors commented and listed other studies reporting the association between the ARBs and cancers (Table 7), it is still unclear whether the ARBs have survival benefit in most cancers. For now it is evident that treatment with an ARB medication does not increase the risk of cancer (FDA report https://www.fda.gov/drugs/drug-safety-and-availability/fda-drug-safety-communication-no-increase-risk-cancer-certain-blood-pressure-drugs-angiotensin) .

Even though they found through the skilled analysis from their propensity-score-matched cohort, their findings are the association between ARBs use and survival in OSCC patients NOT the causal relationship. So they may not conclude “ARBs have survival benefit in OSCC patients”. It should be very prudent to this conclusion. In this regard, I recommend to erase the final parenthesis “These findings highlight the clinical usefulness of ARBs in OSCC patients with advanced disease” in conclusion. And the authors had better to add some comments on this critical issue regarding the causal relationship.

Reviewer #2: The authors described that long use of ARB is associated with an increased survival

rate and is a positive, independent prognostic factor in patients with oral squamous cell carcinoma (OSCC).

The point of this observation is very interesting, and while the manuscript is well written and are well

performed, I want you to revise the manuscript because of the following questions.

Major questions

1. There is no data of blood pressure. Whether the patients take ARB or not, whether the patients have

hypertension or not, whether the patients have cancer or not, hypertension is the very important

factor of mortality. Therefore, the data of blood pressure of two groups is very important and if

authors prove the usefulness of ARB in the context of increasing survival rate, the authors must

show the data of blood pressure and when the authors analyze the data, the authors must correct

the data of blood pressure.

2. The authors reveal that all patients using ARB have hypertension, but only 37.5 % of the patients not

using ARB have hypertension. If the authors discuss the usefulness of ARB in the context of increasing

survival rate of OSCC patients, the authors need to demonstrate whether the reason of the usefulness

of ARB is class effect or antihypertensive effect.

3. There are various pathological types in oral cancer. Why do the authors choose only squamous cell

carcinoma? Why do the authors exclude non-squamous cell carcinoma during the first stage? I can

understand that the authors analyze by type of pathological findings, however I cannot understand

excluding the patient of non-squamous cell carcinoma. I think the authors must also analyze for the

patients of non-squamous cell carcinoma.

4. Can’t the cancer duration at the start of medication of ARB or antihypertensive agents be a

confounding factor?

5. Why do the authors decide the cut off period of receiving ARB to more than 180 days or not?

6. PLOS authors have the option to publish the peer review history of their article (what does this mean?). If published, this will include your full peer review and any attached files.

Reviewer #1: No

Reviewer #2: No

---

## [Author Response · Author response to Decision Letter 0]

13 Oct 2021

RE: “Angiotensin II receptor blockers and oral squamous cell carcinoma survival: A propensity-score-matched cohort study.”

We would like to thank the reviewers for the thorough reading of our manuscript as well as their valuable comments. We have followed their comments closely and feel that their suggestions have further strengthened the manuscript. Below are our point-by-point responses.

Response to Reviewer’s Comments:

Review 1:

The authors reported that ARB use for more than 180 days is associated with an increased survival rate and is a positive, independent prognostic factor in patients with oral squamous cell carcinoma (OSCC) in their propensity-score-matched cohort from the registry database. It is interesting but also arguable results. As the authors commented and listed other studies reporting the association between the ARBs and cancers (Table 7), it is still unclear whether the ARBs have a survival benefit in most cancers. For now, it is evident that treatment with an ARB medication does not increase the risk of cancer (FDA report https://www.fda.gov/drugs/drug-safety-and-availability/fda-drug-safety-communication-no-increase-risk-cancer-certain-blood-pressure-drugs-angiotensin). Even though they found through the skilled analysis from their propensity-score-matched cohort, their findings are the association between ARBs use and survival in OSCC patients NOT the causal relationship. So they may not conclude “ARBs have survival benefit in OSCC patients”. It should be very prudent to this conclusion. In this regard, I recommend to erase the final parenthesis “These findings highlight the clinical usefulness of ARBs in OSCC patients with advanced disease” in conclusion. And the authors had better to add some comments on this critical issue regarding the causal relationship.

Response: Dear reviewer, I agree with your precise comment. I deleted the final parenthesis according to your opinion and changed it to “A further two-arm study should be conducted to confirm the clinical usefulness of ARBs in OSCC patients.” (Page 2 Abstract; Page 22 Conclusion). We also added some comments on this issue in the Discussion section as the following: “However, there was no consistent conclusion on whether ARBs have a survival benefit in patients coexisting with malignancies. One of the main reasons was the discrepancy in confounding control across these studies. Unknown confounding and selection bias might exist in these retrospective studies with different study design, indicating the findings are only the association between ARBs use and survival. Future two-arm controlled study should be conducted for strengthening the causal relationship.” (Page 17)

 

Review 2:

1. There is no data of blood pressure. Whether the patients take ARB or not, whether the patients have hypertension or not, whether the patients have cancer or not, hypertension is the very important factor of mortality. Therefore, the data of blood pressure of two groups is very important and if authors prove the usefulness of ARB in the context of increasing survival rate, the authors must show the data of blood pressure and when the authors analyze the data, the authors must correct the data of blood pressure.

Response: Dear reviewer, I agree with your opinion very well, and I think what you are concerned about is very important. However, I am sorry that we didn’t have the complete blood pressure (BP) data for each person in our database. The most representative BP comes from the daily measurement by the patient at home, or the BP measured in the clinic at least. However, we only had the BP measured during the hospitalization, which was available for 616 patients. The most crucial limitation from the hospitalization BP was the incomplete data for all patients and many confounding factors existing in this data. For example, patients staying in the hospital might suffer from acute disease, which induced unstable BP measurement. Some patients might even be in a critical status, which caused a lower BP result. Because of the reasons above, I deem it as a major limitation in our study, and put it into our discussion. I hope you could accept my explanation.

 We agree your comment is crucial for future study. Therefore, I analyzed the hospitalization BP regardless of the reasons we mentioned above. The data units of systemic blood pressure (SBP) level below 80 mmHg were deleted first, which I deemed probable data of critical status. Then I collected and averaged all the SBP data for each person (n=616). Afterward, the SBP data were classified into normal and abnormal by BP=140 mmHg. We put the variate into our Cox regression model and found no statistical effect on overall survival (HR high BP vs. normal BP = 0.79, 95% CI = 0.54-1.14). A similar result was found if we changed the divide by BP=120 mmHg (HR high BP vs. normal BP = 0.73, 95% CI = 0.48-1.11). However, we can’t jump to the conclusion that BP was not associated with survival because the above analysis was based on an incomplete and unrepresentative hospitalization BP database. 

 In conclusion, a complete representative BP measurement should be taken into account in a future study. Therefore, I summarized the above explanation and added it into the discussion section: “Lastly, blood pressure, a crucial variate in an anti-hypertensive medication study, was not taken into account in our research. The main reasons were that complete data of blood pressure was not available in our database, and many confounding factors existed in the residual database as well. However, taking representative blood pressure data into analysis should be considered in future studies.” (Page 21)

2. The authors reveal that all patients using ARB have hypertension, but only 37.5 % of the patients not using ARB have hypertension. If the authors discuss the usefulness of ARB in the context of increasing survival rate of OSCC patients, the authors need to demonstrate whether the reason of the usefulness of ARB is class effect or antihypertensive effect.

Response: Dear reviewer, your comment about survival benefit might be from ARB use probably from the antihypertensive effect instead of the class effect. It is a question not easy to answer. Here, I assumed that the survival of patients with ARB use would not be superior to those without hypertension if the antihypertensive effect plays a significant role in survival. This assumption is based on the previous research (Ref 1), in which patients with hypertension had poor long-term survival than patients without hypertension. In other words, the class effect of ARBs on the survival benefit in oral cancer patients could be approved if the survival of patients with ARB use (hypertension group) is superior to those without hypertension.

We tested this hypothesis and calculated the crude and adjusted hazard ratio, including a 95% confidence interval for overall survival in patients with ARBs use (n=357) relative to patients without hypertension (n=223). Notably, 180-day ARB use was consistently associated with improved OS with or without adjustment. (Crude HR ARB users vs. non-HTN = 0.70, 95% CI = 0.50-0.99; adjusted HR ARB users vs. non-HTN = 0.67, 95% CI = 0.48-0.95). The above findings displayed the possible anti-cancer effect of ARBs in oral cancer patients. In addition, there is no evidence that other antihypertensive agents suppress cancer development or progression, which had been described in the Limitation section. The class effect of ARBs was therefore indirectly approved.

We appreciate your incisive comments. The above statements were concisely added into our Discussion section as the following:” Some might argue that the survival benefit from ARB use was probably from the antihypertensive effect instead of the anti-cancer effects. According to a previous study, patients with hypertension had poor long-term survival than those without hypertension [28]. However, in our study, 180-day ARB use in patients with hypertension was consistently associated with improved OS compared to patients without hypertension. (Crude HRARB users vs. non-HTN = 0.70, 95% CI = 0.50-0.99; adjusted HRARB users vs. non-HTN = 0.67, 95% CI = 0.48-0.95). The above findings displayed the possible anti-cancer effect of ARBs in oral cancer patients.” (Page 19)

Ref 1. Andersson OK, Almgren T, Persson B, Samuelsson O, Hedner T, Wilhelmsen L. Survival in treated hypertension: follow up study after two decades. BMJ. 1998;317(7152):167-171. doi:10.1136/bmj.317.7152.167

3. There are various pathological types of oral cancer. Why do the authors choose only squamous cell carcinoma? Why do the authors exclude non-squamous cell carcinoma during the first stage? I can understand that the authors analyze by type of pathological findings. However, I cannot understand excluding the patient of non-squamous cell carcinoma. I think the authors must also analyze for the patients of non-squamous cell carcinoma.

Response: Dear reviewer, what you had concerned about is reasonable. However, according to the past epidemiology study of oral cancer in Asia, the majority (84-97%) of oral cancer is squamous cell carcinoma (Ref 1). One native study in Taiwan also showed that squamous cell carcinoma type occupied 93.1% of oral cancer (Ref 2). Because of the epidemiologic characteristics in Taiwan, we focused on the major squamous cell carcinoma type to prevent accidental confounding from other types. I hope you can accept this explanation.

Ref 1. Krishna Rao SV, Mejia G, Roberts-Thomson K, Logan R. Epidemiology of oral cancer in Asia in the past decade--an update (2000-2012). Asian Pac J Cancer Prev. 2013;14(10):5567-77. doi: 10.7314/apjcp.2013.14.10.5567. 

Ref 2. Huang CC, Ou CY, Lee WT, Hsiao JR, Tsai ST, Wang JD. Life expectancy and expected years of life lost to oral cancer in Taiwan: a nation-wide analysis of 22,024 cases followed for 10 years. Oral Oncol. 2015 Apr;51(4):349-54. doi: 10.1016/j.oraloncology.2015.01.001. Epub 2015 Jan 17. 

4. Can’t the cancer duration at the start of medication of ARB or antihypertensive agents be a confounding factor?

Response: Yes, we agree with your opinion. The duration between initial cancer diagnosis and the start point of ARB use could be a confounding factor, which was often considered an immortal time bias. To prevent such a bias, we had considered “the diagnostic year of oral cancer” as a covariate in both groups when performing propensity score matching. We had mentioned this part in the “Statistical Analyses” section as the following:” Propensity scores were calculated using a logistic regression model with sex, age, pathological AJCC stage, comorbidities, and the diagnostic year of OSCC as covariates (S1 Table). … Because the diagnostic years were matched in both groups, the calculated survival time in non-users started from the same day as its match to make the comparison between the two groups fair.” (Page 6 Statistical Analyses)

5. Why do the authors decide the cut off period of receiving ARB to more than 180 days or not?

Response: Thank you for the excellent comment. Why we decided the cut-off period of 180 days was based on the previous research. The main reason is to control for protopathic bias, and therefore some studies have incorporated the concept of lag-time into their exposure definition. The best lag-time to be applied was also studied, which around six months were the most appropriate period for the assessment of drug exposure (Ref 1). We didn’t explain it clearly in the original manuscript, and therefore, we now added the supplement as the followings: “In addition, the best lag-time to be applied in studies accordingly was around six months, which was the most appropriate period for the assessment of drug exposure [29].” (Page 20)

There was another direct and practical reason. We had run a pre-analysis Kaplan-Meier Method, which considers the possible effects of at least three months (90days) or six months (180 days). The figures are shown below, and the preliminary result showed that only receiving ARBs for more than 180 days had a statistical survival benefit.

Ref 1. Tamim H, Monfared AA, LeLorier J. Application of lag-time into exposure definitions to control protopathic bias. Pharmacoepidemiol Drug Saf. 2007 Mar;16(3):250-8. doi: 10.1002/pds.1360. PMID: 17245804.

---

## [Decision Letter · Decision Letter 1]

3 Nov 2021

PONE-D-21-16892R1Angiotensin II receptor blockers and oral squamous cell carcinoma survival: A propensity-score-matched cohort studyPLOS ONE

Dear Dr. Chiu,

Thank you for submitting your manuscript to PLOS ONE. After careful consideration, we feel that it has merit but does not fully meet PLOS ONE’s publication criteria as it currently stands. Therefore, we invite you to submit a revised version of the manuscript that addresses the points raised during the review process.

The revised manuscript is well assessed by the two reviewers; however, several minor revisions are still necessary in the present form.

See the reviewer's comments carefully and respond them appropriately.

We look forward to receiving your revised manuscript.

Kind regards,

Masaki Mogi

Academic Editor

PLOS ONE

Journal Requirements:

Reviewers' comments:

Reviewer's Responses to Questions

**Comments to the Author**

1. If the authors have adequately addressed your comments raised in a previous round of review and you feel that this manuscript is now acceptable for publication, you may indicate that here to bypass the “Comments to the Author” section, enter your conflict of interest statement in the “Confidential to Editor” section, and submit your "Accept" recommendation.

Reviewer #1: All comments have been addressed

Reviewer #2: (No Response)

2. Is the manuscript technically sound, and do the data support the conclusions?

Reviewer #1: Yes

Reviewer #2: Yes

3. Has the statistical analysis been performed appropriately and rigorously? 

Reviewer #1: Yes

Reviewer #2: Yes

4. Have the authors made all data underlying the findings in their manuscript fully available?

Reviewer #1: Yes

Reviewer #2: Yes

5. Is the manuscript presented in an intelligible fashion and written in standard English?

Reviewer #1: Yes

Reviewer #2: Yes

6. Review Comments to the Author

Reviewer #1: The authors addressed adequately my comments. I think the paper revised properly with adequate answers to reviewer's comments.

Reviewer #2: Thank you for your response to my question.

Most of your response of the authors can be understanded, however, some responses seem to be arguable.

About my question No.1-2, your responses are very well-formed, and your analysis about 616 patients of available information of BP measurements is very important data, however it seems that the reason of survival benefit of using ARB in oral cancer patients is not always anti-cancer effect of ARBs. Even if the effect of survival benefit of using ARB is not dependent on blood pressure, the reasons of survival benefit of using ARB are thought of as several organ protective effect, such as cardioprotective effect or renal protective effect. Therefore, the author should delete enrollment of possible anti-cancer effect of ARBs, and should describe another consideration about survival benefit of using ARB in oral cancer patients.

7. PLOS authors have the option to publish the peer review history of their article (what does this mean?). If published, this will include your full peer review and any attached files.

Reviewer #1: No

Reviewer #2: No

---

## [Author Response · Author response to Decision Letter 1]

8 Nov 2021

RE: “Angiotensin II receptor blockers and oral squamous cell carcinoma survival: A propensity-score-matched cohort study.”

We would like to thank the reviewers for the thorough reading of our manuscript as well as their valuable comments. We have followed their comments closely and feel that their suggestions have further strengthened the manuscript. Below are our point-by-point responses.

Response to Reviewer’s Comments:

Review 1:

The authors addressed adequately my comments. I think the paper revised properly with adequate answers to reviewer's comments.

Response: Dear reviewer, we thanked again for your thorough reading of our manuscript as well as the valuable comments.

Review 2:

Most of your response of the authors can be understanded, however, some responses seem to be arguable. About my question No.1-2, your responses are very well-formed, and your analysis about 616 patients of available information of BP measurements is very important data, however it seems that the reason of survival benefit of using ARB in oral cancer patients is not always anti-cancer effect of ARBs. Even if the effect of survival benefit of using ARB is not dependent on blood pressure, the reasons of survival benefit of using ARB are thought of as several organ protective effect, such as cardioprotective effect or renal protective effect. Therefore, the author should delete enrollment of possible anti-cancer effect of ARBs, and should describe another consideration about survival benefit of using ARB in oral cancer patients.

Response: Dear reviewer, I agree with your opinion addressed” Even if the effect of survival benefit of using ARB is not dependent on blood pressure, the reasons of survival benefit of using ARB are thought of as several organ protective effect, such as cardioprotective effect or renal protective effect.” I think what you mentioned explained the main findings in our result, which the survival benefit only showed statistically significant in OS, but not in DSS. Therefore, some discussion and reference about this issue were added accordingly.

 However, directly deleting possible anti-cancer effect of ARBs concerns me. In our discussion, the evidence of anti-cancer mechanism did exist in the molecular level in head and neck cancer (Ref 1,2). In addition, a newly published article in Cancer (Ref 3) also proved that ARB use provided clinical survival benefits (both OS and DSS) with specific anti-tumor molecular signaling in nasopharyngeal carcinoma, which indicated that ARB might have anti-cancer effects on HNSCC. In our article, the anti-cancer effect didn’t reach a statistically significance (DSS, HRARB users vs. non-users = 0.73, 95% CI = 0.48–1.10), but the effect size was nevertheless similar to OS (HRARB users vs. non-users = 0.73, 95% CI = 0.53-0.99). I would rather interpret that there might probably exist an anti-cancer effect, yet with more variability, which the conclusion of anti-cancer effect can’t be reached in our study. 

Therefore, I changed to use more conservative wording in the text, deleted some arbitrary sentences, and revised relevant reference of anti-cancer in HNSCC. I consider that making a balanced discussion between the effects of ARBs would be better than presenting only one side. I hope you could accept my explanation.

The followings are some changes made in our text:

1. We deleted the sentence” The effectiveness of anti-cancer therapies has been supported in many clinical studies.” (Page 17 Discussion).

2. We deleted the sentence” The results of our clinical study particularly highlight the anti-cancer effects on HNSCC;” (Page 19 Discussion).

3. We deleted the sentence” the ACEI perindopril has been shown to reduce the growth of head and neck squamous cell carcinoma (HNSCC) in vivo [26], suggesting a role for Ang II in HNSCC.” (Page 19 Discussion).

4. We deleted the sentences” Some might argue that the survival benefit from ARB use was probably from the anti-hypertensive effect instead of the anti-cancer effects. According to a previous study, patients with hypertension had poor long-term survival than those without hypertension [31]. However, in our study, 180-day ARB use in patients with hypertension was consistently associated with improved OS compared to patients without hypertension. (Crude HRARB users vs. non-HTN = 0.70, 95% CI = 0.50-0.99; adjusted HRARB users vs. non-HTN = 0.67, 95% CI = 0.48-0.95). The above findings displayed the possible anti-cancer effect of ARBs in oral cancer patients.” (Page 20 Discussion).

5. We balanced the discussion as followings: “…This was approved by Lin et al., which ARBs were found to exert antiproliferative and antiangiogenesis effects by inducing apoptosis in nasopharyngeal carcinoma (NPC). In addition, improved 5-year OS and DSS were found among patients with NPC using ARBs [27]. Interestingly, we found that ARB use for at least 180 days improved the OS rate of patients with OSCC statistically, yet not DSS rate (Table 5). Accordingly, RAS is a major regulator of blood pressure (BP) and vascular response to injury. There is large evidence that RAS inhibition provides end-organ protection independent of BP lowering [28]. That probably explained the survival benefit of using ARBs is mainly through as end-organ protective effect, which further reduced overall mortality [29].As for the anti-cancer effect of ARBs in oral cancer patients, it still remained controversial in our current study.” (Page 19 Discussion).

Ref 1. George, A.J., W.G. Thomas, and R.D. Hannan, The renin-angiotensin system and cancer: old dog, new tricks. Nat Rev Cancer, 2010. 10(11): p. 745-59.

Ref 2. Hinsley, E.E., et al., Angiotensin 1-7 inhibits angiotensin II-stimulated head and neck cancer progression. Eur J Oral Sci, 2017. 125(4): p. 247-257.

Ref 3. Lin, Y.-T., et al., Angiotensin II receptor blockers valsartan and losartan improve survival rate clinically and suppress tumor growth via apoptosis related to PI3K/AKT signaling in nasopharyngeal carcinoma. Cancer, 2021. 127(10): p. 1606-1619.

---

## [Decision Letter · Decision Letter 2]

17 Nov 2021

Angiotensin II receptor blockers and oral squamous cell carcinoma survival: A propensity-score-matched cohort study

PONE-D-21-16892R2

Dear Dr. Chiu,

We’re pleased to inform you that your manuscript has been judged scientifically suitable for publication and will be formally accepted for publication once it meets all outstanding technical requirements.

Kind regards,

Masaki Mogi

Academic Editor

PLOS ONE

Additional Editor Comments (optional):

Reviewers' comments:

Reviewer's Responses to Questions

**Comments to the Author**

1. If the authors have adequately addressed your comments raised in a previous round of review and you feel that this manuscript is now acceptable for publication, you may indicate that here to bypass the “Comments to the Author” section, enter your conflict of interest statement in the “Confidential to Editor” section, and submit your "Accept" recommendation.

Reviewer #2: All comments have been addressed

2. Is the manuscript technically sound, and do the data support the conclusions?

Reviewer #2: Yes

3. Has the statistical analysis been performed appropriately and rigorously? 

Reviewer #2: Yes

4. Have the authors made all data underlying the findings in their manuscript fully available?

Reviewer #2: Yes

5. Is the manuscript presented in an intelligible fashion and written in standard English?

Reviewer #2: Yes

6. Review Comments to the Author

Reviewer #2: (No Response)

7. PLOS authors have the option to publish the peer review history of their article (what does this mean?). If published, this will include your full peer review and any attached files.

Reviewer #2: No

---

## [Editor Report · Acceptance letter]

22 Nov 2021

PONE-D-21-16892R2 

Angiotensin II receptor blockers and oral squamous cell carcinoma survival: A propensity-score-matched cohort study 

Dear Dr. Chiu:

I'm pleased to inform you that your manuscript has been deemed suitable for publication in PLOS ONE. Congratulations! Your manuscript is now with our production department. 

Kind regards, 

on behalf of

Dr. Masaki Mogi 

Academic Editor

PLOS ONE